# Unsupervised Learning of Disentangled Representations from Video

**Emily Denton**
Department of Computer Science
New York University
denton@cs.nyu.edu

**Vighnesh Birodkar**
Department of Computer Science
New York University
vighneshbirodkar@nyu.edu

## Abstract

We present a new model DRNET that learns disentangled image representations from video. Our approach leverages the temporal coherence of video and a novel adversarial loss to learn a representation that factorizes each frame into a stationary part and a temporally varying component. The disentangled representation can be used for a range of tasks. For example, applying a standard LSTM to the time-vary components enables prediction of future frames. We evaluate our approach on a range of synthetic and real videos, demonstrating the ability to coherently generate hundreds of steps into the future.

## 1 Introduction

Unsupervised learning from video is a long-standing problem in computer vision and machine learning. The goal is to learn, without explicit labels, a representation that generalizes effectively to a previously unseen range of tasks, such as semantic classification of the objects present, predicting future frames of the video or classifying the dynamic activity taking place. There are several prevailing paradigms: the first, known as self-supervision, uses domain knowledge to implicitly provide labels (e.g. predicting the relative position of patches on an object [4] or using feature tracks [36]). This allows the problem to be posed as a classification task with self-generated labels. The second general approach relies on auxiliary action labels, available in real or simulated robotic environments. These can either be used to train action-conditional predictive models of future frames [2, 20] or inverse-kinematics models [1] which attempt to predict actions from current and future frame pairs. The third and most general approaches are predictive auto-encoders (e.g.[11, 12, 18, 31]) which attempt to predict future frames from current ones. To learn effective representations, some kind of constraint on the latent representation is required.

In this paper, we introduce a form of predictive auto-encoder which uses a novel adversarial loss to factor the latent representation for each video frame into two components, one that is roughly time-independent (i.e. approximately constant throughout the clip) and another that captures the dynamic aspects of the sequence, thus varying over time. We refer to these as *content* and *pose* components, respectively. The adversarial loss relies on the intuition that while the content features should be distinctive of a given clip, individual pose features should not. Thus the loss encourages pose features to carry no information about clip identity. Empirically, we find that training with this loss to be crucial to inducing the desired factorization.

We explore the disentangled representation produced by our model, which we call Disentangled-Representation Net (DRNET ), on a variety of tasks. The first of these is predicting future video frames, something that is straightforward to do using our representation. We apply a standard LSTM model to the pose features, conditioning on the content features from the last observed frame. Despite the simplicity of our model relative to other video generation techniques, we are able to generate convincing long-range frame predictions, out to hundreds of time steps in some instances. This is significantly further than existing approaches that use real video data. We also show that DRNET can

be used for classification. The content features capture the semantic content of the video thus can be used to predict object identity. Alternately, the pose features can be used for action prediction.

## 2    Related work

On account of its natural invariances, image data naturally lends itself to an explicit "what" and "where" representation. The capsule model of Hinton *et al.* [10] performed this separation via an explicit auto-encoder structure. Zhao *et al.* [40] proposed a multi-layered version, which has similarities to ladder networks [23]. Several weakly supervised approaches have been proposed to factor images into style and content (e.g. [19, 24]). These methods all operate on static images, whereas our approach uses temporal structure to separate the components.

Factoring video into time-varying and time-independent components has been explored in many settings. Classic structure-from-motion methods use an explicit affine projection model to extract a 3D point cloud and camera homography matrices [8]. In contrast, Slow Feature Analysis [38] has no model, instead simply penalizing the rate of change in time-independent components and encouraging their decorrelation. Most closely related to ours is Villegas *et al.* [33] which uses an unsupervised approach to factoring video into content and motion. Their architecture is also broadly similar to ours, but the loss functions differ in important ways. They rely on pixel/gradient space $\ell_p$-norm reconstructions, plus a GAN term [6] that encourages the generated frames to be sharp. We also use an $\ell_2$ pixel-space reconstruction. However, this pixel-space loss is only applied, in combination with a novel adversarial term applied to the pose features, to learn the disentangled representation. In contrast to [33], our forward model acts on latent pose vectors rather than predicting pixels directly.

Other approaches explore general methods for learning disentangled representations from video. Kulkarni *et al.* [14] show how explicit graphics code can be learned from datasets with systematic dimensions of variation. Whitney *et al.* [37] use a gating principle to encourage each dimension of the latent representation to capture a distinct mode of variation. Grathwohl *et al.* [7] propose a deep variational model to disentangle space and time in video sequences.

A range of generative video models, based on deep nets, have recently been proposed. Ranzato *et al.* [22] adopt a discrete vector quantization approach inspired by text models. Srivastava *et al.* [31] use LSTMs to generate entire frames. Video Pixel Networks [12] use these models is a conditional manner, generating one pixel at a time in raster-scan order (similar image models include [27, 32]). Finn *et al.* [5] use an LSTM framework to model motion via transformations of groups of pixels. Cricri *et al.* [3] use a ladder of stacked-autoencoders. Other works predict optical flows fields that can be used to extrapolate motion beyond the current frame, e.g. [17, 39, 35]. In contrast, a single pose vector is predicted in our model, rather than a spatial field.

Chiappa *et al.* [2] and Oh *et al.* [20] focus on prediction in video game environments, where known actions at each frame can be permit action-conditional generative models that can give accurate long-range predictions. In contrast to the above works, whose latent representations combine both content and motion, our approach relies on a factorization of the two, with a predictive model only being applied to the latter. Furthermore, we do not attempt to predict pixels directly, instead applying the forward model in the latent space. Chiappa *et al.* [2], like our approach, produces convincing long-range generations. However, the video game environment is somewhat more constrained than the real-world video we consider since actions are provided during generation.

Several video prediction approaches have been proposed that focus on handling the inherent uncertainty in predicting the future. Mathieu *et al.* [18] demonstrate that a loss based on GANs can produce sharper generations than traditional $\ell_2$-based losses. [34] train a series of models, which aim to span possible outcomes and select the most likely one at any given instant. While we considered GAN-based losses, the more constrained nature of our model, and the fact that our forward model does not directly generate in pixel-space, meant that standard deterministic losses worked satisfactorily.

## 3    Approach

In our model, two separate encoders produce distinct feature representations of content and pose for each frame. They are trained by requiring that the content representation of frame $x^t$ and the pose representation of future frame $x^{t+k}$ can be combined (via concatenation) and decoded to predict the pixels of future frame $x^{t+k}$. However, this reconstruction constraint alone is insufficient to induce

the desired factorization between the two encoders. We thus introduce a novel adversarial loss on the pose features that prevents them from being discriminable from one video to another, thus ensuring that they cannot contain content information. A further constraint, motivated by the notion that content information should vary slowly over time, encourages temporally close content vectors to be similar to one another.

More formally, let $x_i = (x_i^1, ..., x_i^T)$ denote a sequence of $T$ images from video $i$. We subsequently drop index $i$ for brevity. Let $E_c$ denote a neural network that maps an image $x^t$ to the *content* representation $h_c^t$ which captures structure shared across time. Let $E_p$ denote a neural network that maps an image $x^t$ to the *pose* representation $h_p^t$ capturing content that varies over time. Let $D$ denote a decoder network that maps a content representation from a frame, $h_c^t$, and a pose representation $h_p^{t+k}$ from future time step $t + k$ to a prediction of the future frame $\tilde{x}^{t+k}$. Finally, $C$ is the *scene discriminator network* that takes pairs of pose vectors $(h_1, h_2)$ and outputs a scalar probability that they came from the same video or not.

The loss function used during training has several terms:

**Reconstruction loss:** We use a standard per-pixel $\ell_2$ loss between the predicted future frame $\tilde{x}^{t+k}$ and the actual future frame $x^{t+k}$ for some random frame offset $k \in [0, K]$:

$$\mathcal{L}_{reconstruction}(D) = ||D(h_c^t, h_p^{t+k}) - x^{t+k}||_2^2 \tag{1}$$

Note that many recent works on video prediction that rely on more complex losses that can capture uncertainty, such as GANs [19, 6].

**Similarity loss:** To ensure the content encoder extracts mostly time-invariant representations, we penalize the squared error between the content features $h_c^t, h_c^{t+k}$ of neighboring frames $k \in [0, K]$:

$$\mathcal{L}_{similarity}(E_c) = ||E_c(x^t) - E_c(x^{t+k})||_2^2 \tag{2}$$

**Adversarial loss:** We now introduce a novel adversarial loss that exploits the fact that the objects present do not typically change *within* a video, but they do *between* different videos. Our desired disenanglement would thus have the content features be (roughly) constant within a clip, but distinct between them. This implies that the pose features should not carry any information about the identity of objects within a clip.

We impose this via an adversarial framework between the scene discriminator network $C$ and pose encoder $E_p$, shown in Fig. 1. The latter provides pairs of pose vectors, either computed from the same video $(h_{p,i}^t, h_{p,i}^{t+k})$ or from different ones $(h_{p,i}^t, h_{p,j}^{t+k})$, for some other video $j$. The discriminator then attempts to classify the pair as being from the same/different video using a cross-entropy loss:

$$-\mathcal{L}_{adversarial}(C) = \log(C(E_p(x_i^t), E_p(x_i^{t+k}))) + \log(1 - C(E_p(x_i^t), E_p(x_j^{t+k}))) \tag{3}$$

The other half of the adversarial framework imposes a loss function on the pose encoder $E_p$ that tries to maximize the uncertainty (entropy) of the discriminator output on pairs of frames from the same clip:

$$-\mathcal{L}_{adversarial}(E_p) = \frac{1}{2}\log(C(E_p(x_i^t), E_p(x_i^{t+k}))) + \frac{1}{2}\log(1 - C(E_p(x_i^t), E_p(x_i^{t+k}))) \tag{4}$$

Thus the pose encoder is encouraged to produce features that the discriminator is unable to classify if they come from the same clip or not. In so doing, the pose features cannot carry information about object content, yielding the desired factorization. Note that this does assume that the object's pose is not distinctive to a particular clip. While adversarial training is also used by GANs, our setup purely considers classification; there is no generator network, for example.

**Overall training objective:**
During training we minimize the sum of the above losses, with respect to $E_c, E_p, D$ and $C$:

$$\mathcal{L} = \mathcal{L}_{reconstruction}(E_c, E_p, D) + \alpha\mathcal{L}_{similarity}(E_c) + \beta(\mathcal{L}_{adversarial}(E_p) + \mathcal{L}_{adversarial}(C)) \tag{5}$$

where $\alpha$ and $\beta$ are hyper-parameters. The first three terms can be jointly optimized, but the discriminator $C$ is updated while the other parts of the model ($E_c, E_p, D$) are held constant. The overall model is shown in Fig. 1. Details of the training procedure and model architectures for $E_c, E_p, D$ and $C$ are given in Section 4.1.

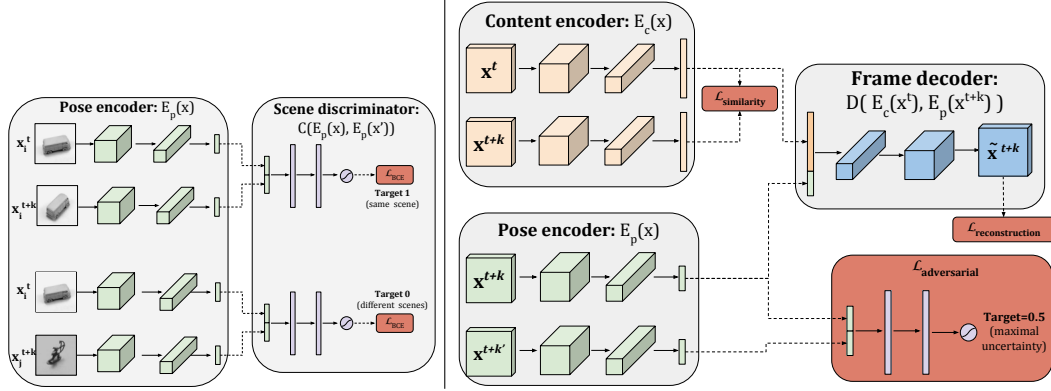

Figure 1: Left: The discriminator $C$ is trained with binary cross entropy (BCE) loss to predict if a pair of pose vectors comes from the same (top portion) or different (lower portion) scenes. $x_i$ and $x_j$ denote frames from different sequences $i$ and $j$. The frame offset $k$ is sampled uniformly in the range $[0, K]$. Note that when $C$ is trained, the pose encoder $E_p$ is fixed. Right: The overall model, showing all terms in the loss function. Note that when the pose encoder $E_p$ is updated, the scene discriminator is held fixed.

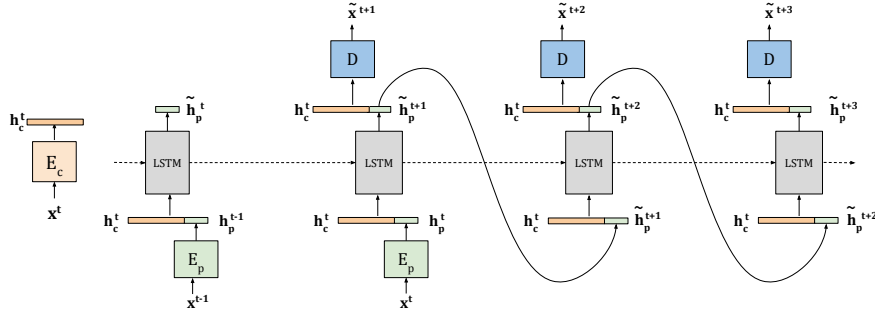

Figure 2: Generating future frames by recurrently predicting $h_p$, the latent pose vector.

## 3.1 Forward Prediction

After training, the pose and content encoders $E_p$ and $E_c$ provide a representation which enables video prediction in a straightforward manner. Given a frame $x^t$, the encoders produce $h_p^t$ and $h_c^t$ respectively. To generate the next frame, we use these as input to an LSTM model to predict the next pose features $h_p^{t+1}$. These are then passed (along with the content features) to the decoder, which generates a pixel-space prediction $\tilde{x}^{t+1}$:

$$\tilde{h}_p^{t+1} = LSTM(E_p(x^t), h_c^t) \qquad \tilde{x}^{t+1} = D(\tilde{h}_p^{t+1}, h_c^t) \qquad (6)$$
$$\tilde{h}_p^{t+2} = LSTM(\tilde{h}_p^{t+1}, h_c^t) \qquad \tilde{x}^{t+2} = D(\tilde{h}_p^{t+2}, h_c^t) \qquad (7)$$

Note that while pose estimates are generated in a recurrent fashion, the content features $h_c^t$ remain fixed from the last observed real frame. This relies on the nature of $\mathcal{L}_{reconstruction}$ which ensured that content features can be combined with future pose vectors to give valid reconstructions.

The LSTM is trained separately from the main model using a standard $\ell_2$ loss between $\tilde{h}_p^{t+1}$ and $h_p^{t+1}$. Note that this generative model is far simpler than many other recent approaches, e.g. [12]. This largely due to the forward model being applied within our disentangled representation, rather than directly on raw pixels.

## 3.2 Classification

Another application of our disentangled representation is to use it for classification tasks. Content features, which are trained to be invariant to local temporal changes, can be used to classify the semantic content of an image. Conversely, a sequence of pose features can be used to classify actions in a video sequence. In either case, we train a two layer classifier network $S$ on top of either $h_c$ or $h_p$, with its output predicting the class label $y$.

# 4   Experiments

We evaluate our model on both synthetic (MNIST, NORB, SUNCG) and real (KTH Actions) video datasets. We explore several tasks with our model: (i) the ability to cleanly factorize into content and pose components; (ii) forward prediction of video frames using the approach from Section 3.1; (iii) using the pose/content features for classification tasks.

## 4.1   Model details

We explored a variety of convolutional architectures for the content encoder $E_c$, pose encoder $E_p$ and decoder $D$. For MNIST, $E_c, E_p$ and $D$ all use a DCGAN architecture [21] with $|h_p| = 5$ and $|h_c| = 128$. The encoders consist of 5 convolutional layers with subsampling. Batch normalization and Leaky ReLU's follow each convolutional layer except the final layer which normalizes the pose/content vectors to have unit norm. The decoder is a mirrored version of the encoder with 5 deconvolutional layers and a sigmoid output layer.

For both NORB and SUNCG, $D$ is a DCGAN architecture while $E_c$ and $E_p$ use a ResNet-18 architecture [9] up until the final pooling layer with $|h_p| = 10$ and $|h_c| = 128$.

For KTH, $E_p$ uses a ResNet-18 architecture with $|h_p| = 24$. $E_c$ uses the same architecture as VGG16 [29] up until the final pooling layer with $|h_c| = 128$. The decoder is a mirrored version of the content encoder with pooling layers replaced with spatial up-sampling. In the style of U-Net [25], we add skip connections from the content encoder to the decoder, enabling the model to easily generate static background features.

In all experiments the scene discriminator $C$ is a fully connected neural network with 2 hidden layers of 100 units. We trained all our models with the ADAM optimizer [13] and learning rate $\eta = 0.002$. We used $\beta = 0.1$ for MNIST, NORB and SUNCG and $\beta = 0.0001$ for KTH experiments. We used $\alpha = 1$ for all datasets.

For future prediction experiments we train a two layer LSTM with 256 cells using the ADAM optimizer. On MNIST, we train the model by observing 5 frames and predicting 10 frames. On KTH, we train the model by observing 10 frames and predicting 10 frames.

## 4.2   Synthetic datasets

**MNIST:** We start with a toy dataset consisting of two MNIST digits bouncing around a 64x64 image. Each video sequence consists of a different pair of digits with independent trajectories. Fig. 3(left) shows how the content vector from one frame and the pose vector from another generate new examples that transfer the content and pose from the original frames. This demonstrates the clean disentanglement produced by our model. Interestingly, for this data we found it to be necessary to use a different color for the two digits. Our adversarial term is so aggressive that it prevents the

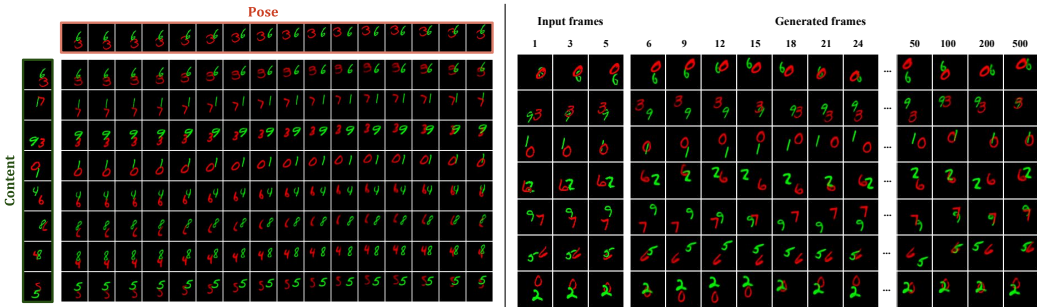

Figure 3: Left: Demonstration of content/pose factorization on held out MNIST examples. Each image in the grid is generated using the pose and content vectors $h_p$ and $h_c$ taken from the corresponding images in the top row and first column respectively. The model has clearly learned to disentangle content and pose. Right: Each row shows forward modeling up to 500 time steps into the future, given 5 initial frames. For each generation, note that only the pose part of the representation is being predicted from the previous time step (using an LSTM), with the content vector being fixed from the 5th frame. The generations remain crisp despite the long-range nature of the predictions.

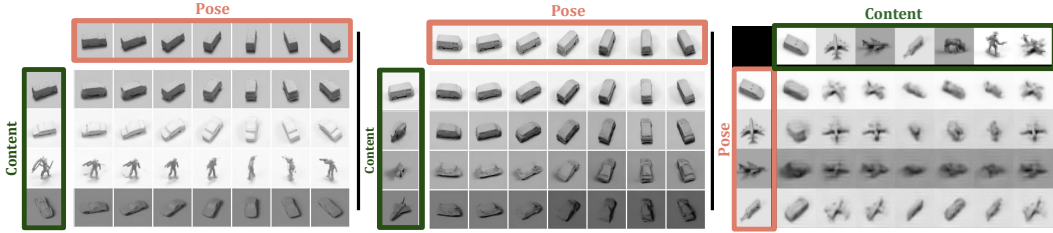

Figure 4: Left: Factorization examples using our DRNET model on held out NORB images. Each image in the grid is generated using the pose and content vectors $h_p$ and $h_c$ taken from the corresponding images in the top row and first column respectively. Center: Examples where DRNET was trained without the adversarial loss term. Note how content and pose are no longer factorized cleanly: the pose vector now contains content information which ends up dominating the generation. Right: factorization examples from Mathieu *et al.* [19].

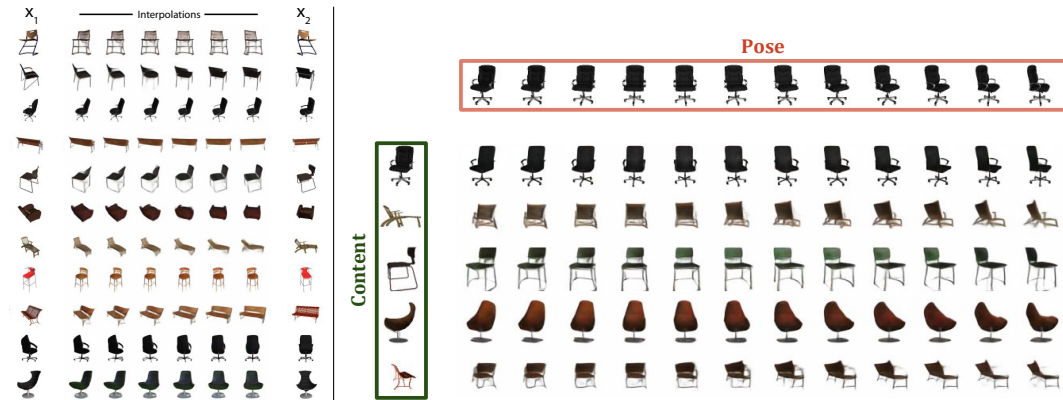

Figure 5: Left: Examples of linear interpolation in pose space between the examples $x_1$ and $x_2$. Right: Factorization examples on held out images from the SUNCG dataset. Each image in the grid is generated using the pose and content vectors $h_p$ and $h_c$ taken from the corresponding images in the top row and first column respectively. Note how, even for complex objects, the model is able to rotate them accurately.

pose vector from capturing any content information, thus without a color cue the model is unable to determine which pose information to associate with which digit. In Fig. 3(right) we perform forward modeling using our representation, demonstrating the ability to generate crisp digits 500 time steps into the future.

**NORB:** We apply our model to the NORB dataset [16], converted into videos by taking sequences of different azimuths, while holding object identity, lighting and elevation constant. Fig. 4.2(left) shows that our model is able to factor content and pose cleanly on held out data. In Fig. 4.2(center) we train a version of our model without the adversarial loss term, which results in a significant degradation in the model and the pose vectors are no longer isolated from content. For comparison, we also show the factorizations produced by Mathieu *et al.* [19], which are less clean, both in terms of disentanglement and generation quality than our approach. Table 1 shows classification results on NORB, following the training of a classifier on pose features and also content features. When the adversarial term is used ($\beta = 0.1$) the content features perform well. Without the term, content features become less effective for classification.

**SUNCG:** We use the rendering engine from the SUNCG dataset [30] to generate sequences where the camera rotates around a range of 3D chair models. The dataset consists of 324 different chair models of varying size, shape and color. DRNET learns a clean factorization of content and pose and is able to generate high quality examples of this dataset, as shown in Fig. 4.2(right).

## 4.3 KTH Action Dataset

Finally, we apply DRNET to the KTH dataset [28]. This is a simple dataset of real-world videos of people performing one of six actions (walking, jogging, running, boxing, handwaving, hand-clapping) against fairly uniform backgrounds. In Fig. 4.3 we show forward generations of different held out examples, comparing against two baselines: (i) the MCNet of Villegas *et al.* [33]which, to the best of our knowledge, produces the current best quality generations of on real-world video and (ii) a baseline auto-encoder LSTM model (AE-LSTM). This is essentially the same as ours, but with a single encoder whose features thus combine content and pose (as opposed to factoring them in DRNET ). It is also similar to [31].

Fig. 7 shows more examples, with generations out to 100 time steps. For most actions this is sufficient time for the person to have left the frame, thus further generations would be of a fixed background. In Fig. 9 we attempt to quantify the fidelity of the generations by comparing our approach to MCNet [33] using a metric derived from the Inception score [26]. The Inception score is used for assessing generations from GANs and is more appropriate for our scenario that traditional metrics such as PSNR or SSIM (see appendix B for further discussion). The curves show the mean scores of our generations decaying more gracefully than MCNet [33]. Further examples and generated movies may be viewed in appendix A and also at `https://sites.google.com/view/drnet-paper//`.

A natural concern with high capacity models is that they might be memorizing the training examples. We probe this in Fig. 4.3, where we show the nearest neighbors to our generated frames from the training set. Fig. 8 uses the pose representation produced by DRNET to train an action classifier from very few examples. We extract pose vectors from video sequences of length 24 and train a fully connected classifier on these vectors to predict the action class. We compare against an autoencoder baseline, which is the same as ours but with a single encoder whose features thus combine content and pose. We find the factorization significantly boosts performance.

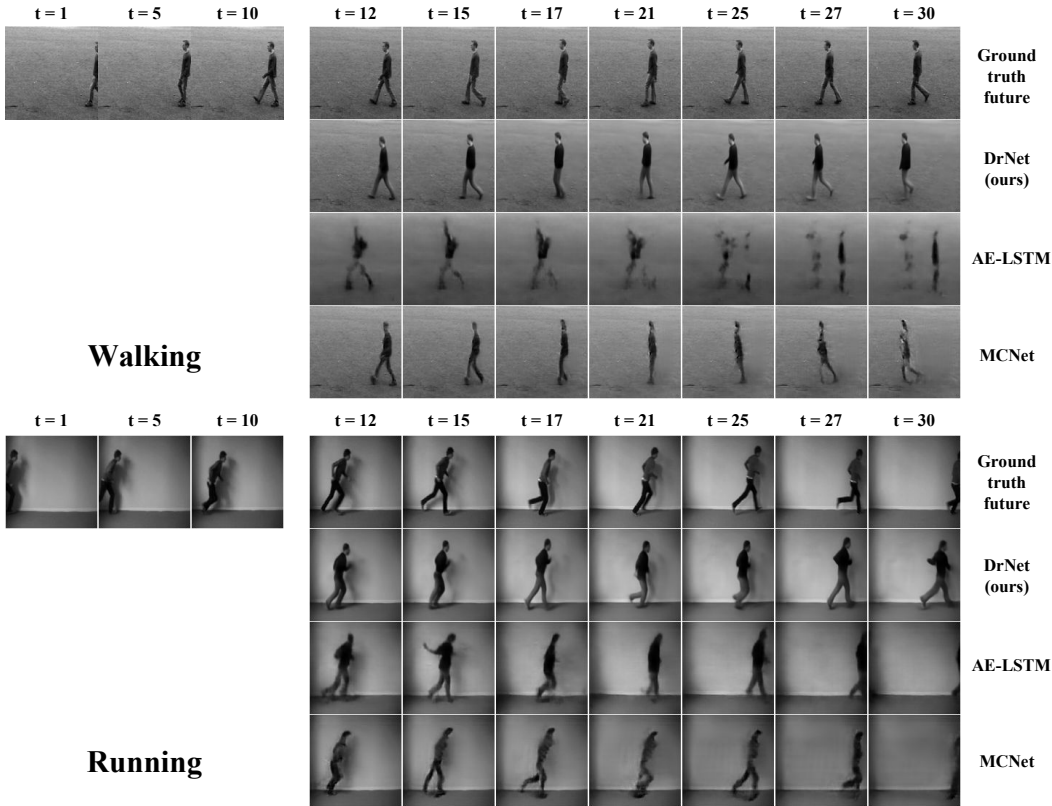

Figure 6: Qualitative comparison between our DRNET model, MCNet [33] and the AE-LSTM baseline. All models are conditioned on the first 10 video frames and generate 20 frames. We display predictions of every $3^{rd}$ frame. Video sequences are taken from held out examples of the KTH dataset for the classes of walking (top) and running (bottom).

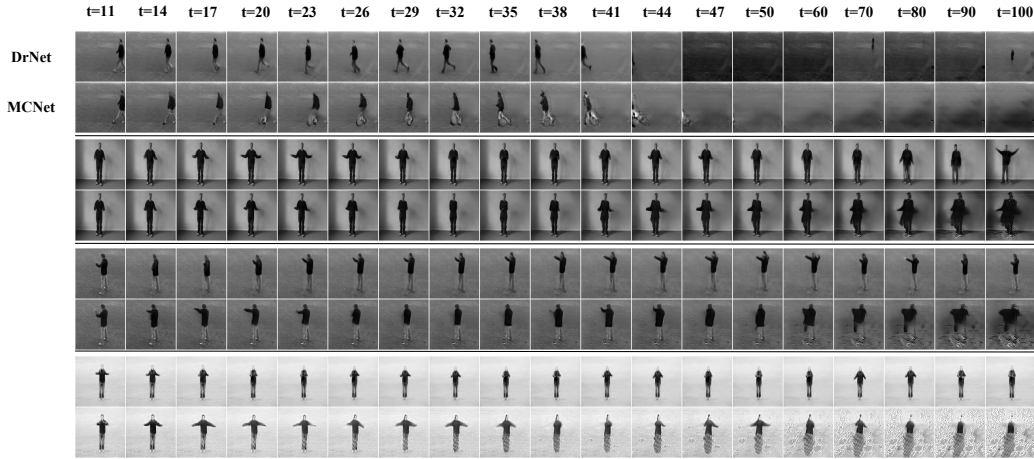

Figure 7: Four additional examples of generations on held out examples of the KTH dataset, rolled out to 100 timesteps.

| Model | | Accuracy (%) |
|---|---|---|
| DRNET $_{\beta=0.1}$ | $h_c$ | **93.3** |
| | $h_p$ | 60.9 |
| DRNET $_{\beta=0}$ | $h_c$ | 72.6 |
| | $h_p$ | 80.8 |
| Mathieu *et al.* [19] | | 86.5 |

Table 1: Classification results on NORB dataset, with/without adversarial loss ($\beta = 0.1/0$) using content or pose representations ($h_c$, $h_p$ respectively). The adversarial term is crucial for forcing semantic information into the content vectors – without it performance drops significantly.

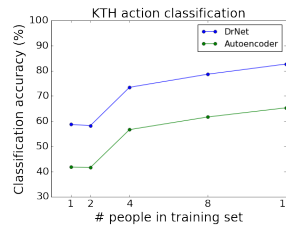

Figure 8: Classification of KTH actions from pose vectors with few labeled examples, with autoencoder baseline. N.B. SOA (fully supervised) is 93.9% [15].

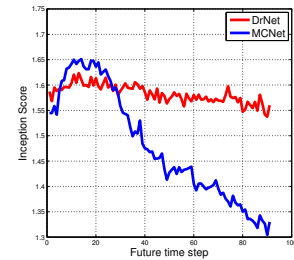

Figure 9: Comparison of KTH video generation quality using Inception score. X-axis indicated how far from conditioned input the start of the generated sequence is.

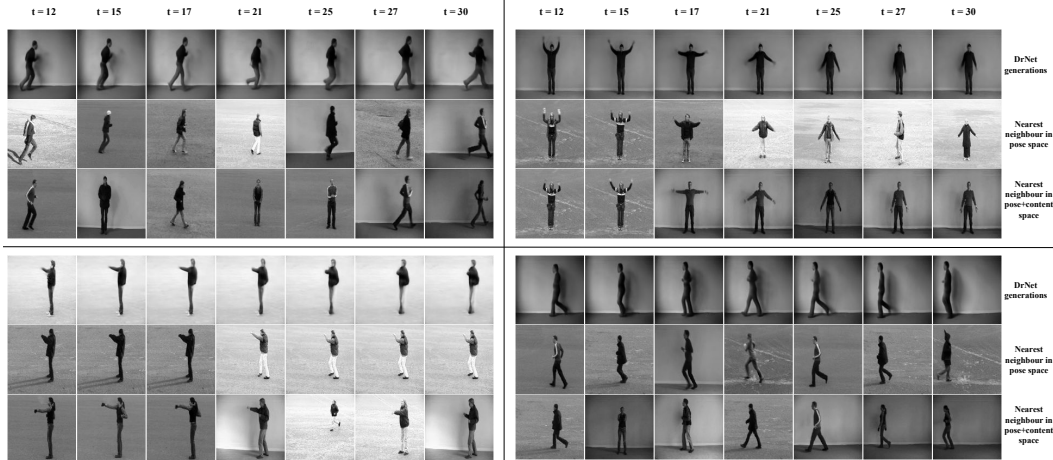

Figure 10: For each frame generated by DRNET (top row in each set), we show nearest-neighbor images from the training set, based on pose vectors (middle row) and both content and pose vectors (bottom row). It is evident that our model is not simply copying examples from the training data. Furthermore, the middle row shows that the pose vector generalizes well, and is independent of background and clothing.

## 5   Discussion

In this paper we introduced a model based on a pair of encoders that factor video into content and pose. This seperation is achieved during training through novel adversarial loss term. The resulting representation is versatile, in particular allowing for stable and coherent long-range prediction through nothing more than a standard LSTM. Our generations compare favorably with leading approaches, despite being a simple model, e.g. lacking the GAN losses or probabilistic formulations of other video generation approaches. Source code is available at `https://github.com/edenton/drnet`.

**Acknowledgments**

We thank Rob Fergus, Will Whitney and Jordan Ash for helpful comments and advice. Emily Denton is grateful for the support of a Google Fellowship

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
