[Supplementary Material · supp.pdf]

# Unsupervised Learning of Disentangled Representations from Video: Supplementary Material

**Emily Denton**
Department of Computer Science
New York University
denton@cs.nyu.edu

**Vighnesh Birodkar**
Department of Computer Science
New York University
vighneshbirodkar@nyu.edu

## A    Further KTH generations

Fig. 1 shows additional long-range KTH sequences generated from our model and MCNet [4]. Generations in movie form are viewable at `https://sites.google.com/view/drnet-paper/`.

## B    Quantitative metrics for evaluating generations

Evaluating samples from generative models is generally problematic. Pixel-wise measures like PNSR and SSIM [5] are appropriate when objects are well aligned, but for long-range generations this is unlikely to be the case. Fig. 2 shows sequences generated from our model and MCNet [4], as well as their difference with respect to the ground truth. While the person remains sharp, there is some error in the velocity prediction, which accumulates to a significant offset in position. Consequently, the resulting PSNR and SSIM scores are very misleading and we adopt the Inception score [2] as an alternative in the main paper.

The Inception Score is computed by first training a classifier network to accurately predict the action class of a video from a sequence of 10 frames. We employ a convolutional network classification architecture where the video frames are concatenated as input to the first layer. Once the classifier is trained, we evaluate the samples generated by DRNET and MCNet by considering the label distribution predicted by the classifier for generated sequences. Intuitively, we expect a good generative model to produce videos with a highly peaked conditional label distribution $p(y|\mathbf{x})$ and a uniform marginal label distribution $p(y)$. Formally, the Inception Score is given by $\exp(\mathbb{E}_x \text{KL}(p(y|\mathbf{x})||p(y))$. Fig. **??** plots the mean Inception Score for generated sequences from our DRNET model and MCnet [4]. The x-axis indicates the offset, from the final frame ofthe conditioned input, of the first generated frame used for the Inception Score.

## C    KTH experimental settings

The KTH dataset consists of 25 different subjects performing six different actions (boxing, hand waving, hand clapping, jogging, running and walking) against a static background. Each person is observed performaing every action in four different scenarios with varied clothing a background conditions. Following [4] we used person 1-16 for training and person 17-25 for testing[1]. We also resize frames to $128{\times}128$ pixels.

# D   Details of classification experiments

**NORB object classification:** We used a two layer fully connected network with 256 hidden units as the classifier. Leaky ReLU's, batch normalization and dropout were used in every layer. We trained with ADAM as used early stopping on a validation set to prevent over fitting.

Figure 1: Further long range generations on KTH dataset. Each pair of rows shows generations by DRNET (top) and MCNet (bottom) for the same conditioning frames.

**KTH action classification:** We used a two layer fully connected network with 1200 hidden units as the classifier. Leaky ReLU's, batch normalization and dropout were used in every layer. For KTH experiments, both DRNET and the autoencoder baseline produced 24 dimensional latent vectors. The classifier was trained on sequences of length 24 so the input to the classifier was $24 \times 24$. We also tried an autoencoder baseline with a 128 dimensional latent space (i.e, same dimensionality as the ceontent vectors of DRNET ) but found this model performed worse. In order to assess the performance of the DRNET and autoencoder representations we trained a classifier for varying training set sizes. Specifically, we varied the number of subjects used in the training set from 1 to 12. We trained with ADAM as used early stopping on a validation set to prevent over fitting.

Figure 2: Comparison of generated sequences from DRNET and MCNet along with corresponding PSNR scores.

## Footnotes

[1]Note, this is not the standard train/test split used for KTH action classification