[Reviews · NeurIPS 2017]

Reviewer 1



This paper presents a neural network architecture and video-based objective function formulation for the disentanglement of pose and content features in each frame. The proposed neural network consists of encoder CNNs and a decoder CNN. The encoder CNNs are trained to decompose frame inputs into contents features (i.e. background, colors of present structures, etc), and pose features (i.e. current configuration of the structures in the video). A decoder CNN is then used to combine the content features of one frame and pose features of a different frame to reconstruct the frame that corresponds to the pose features. This paper presents a new loss for the disentanglement of pose features from input images using a discriminator network (similar to adversarial loss). They train a discriminator network find similarities between features from the pose encoder of images from the same video but different time steps. The pose encoder is then trained to find features that the discriminator cannot distinguish as being similar. At convergence, the pose encoder features will only contain difference between the frames which should reflect the pose change of the person in time. After the encoder-decoder CNN has been trained to identify the content and pose features, an LSTM is trained to take the content features of the last frame and previous pose features to predict the next pose features. The disentangled content and pose features result in high quality video prediction. The experimental results backup the advantage of this method in terms of feature quality for classification tasks, and shows quantitative and qualitative performance boost over the state-of-the-art in video prediction. Pros: Novel video-based loss functions for feature disentanglement. Excellent results on video prediction using the learned pose and content features. Cons: Although this paper claims to be an unsupervised method, the proposed objective function actually needs some weak supervision that different videos should have different content, e.g. object instances; otherwise the second term in equation 3 could be negatively affected (classifying images from different videos as different when they may look the same). The pose LSTM is trained separately from the encoder-decoder CNN. Can the authors comment on why joint end-to-end training was not performed? In fact, similar networks for end-to-end video prediction have been experimented in Oh et al, NIPS 2015. Some typos in equations and text. Should equation 3 be L_{adversarial}(C) instead of L_{adversarial}(D)? h^t_c in LSTM input could be denoted as h^n where n is the last observed frame index, t makes it seem as the current step in generation. Lines 204 and 219 point to Fig. 4.3 which is not in the main paper. Figure 4 should have matching test sequences since it expect us to compare the “left”, “middle”, and “right” results. Action classification experiments are a little confusing: It is not clear what features are used to determine action class and how they are used (i.e. concatenated features? Single features and average for each video? Classification on clips?). The paper states: “In either case, we train a two layer classifier network S on top of either hc or hp, with its output predicting the class label y.” In KTH, the content is not a good indicator about the action that is happening in the video since the same people perform all actions in similar backgrounds. Thus it’s not clear how the content features can be useful for action classification. Classification results of network trained without pose disentanglement loss can make classification results stronger if previous issues are clarified. Pose estimation results using the disentangled pose features are strongly recommended that will significantly strengthen the paper. For the SUNCG dataset (3D object renderings), comparison with the weakly-supervised pose disentangling work by Yang et al, NIPS 2015 is suggested to highlight the performance of the proposed method. The use of the first term in L_{adversarial}(E_p) is not clear to me, it seems that the first term and second term in this loss are pulling the features in completely opposite directions which, to my understanding, is helping D instead of confusing it. Isn’t confusing D the whole purpose of L_{adversarial}(E_p) to confuse D? Title can be more specific to the contribution (adversarial-like loss for content-pose feature disentangling)

Reviewer 2



This paper proposes a new video prediction model called DrNet that achieves very compelling long-term video prediction (on the order of hundreds of frames) by learning a disentangled feature space using both pixel prediction and adversarial losses on the latent space. The proposed model uses a similar convolutional LSTM setup with content and pose factors as in previous video prediction models. The novel aspect is the introduction of a scene discriminator and a pose encoder network. The scene discriminator is trained to distinguish whether two pose encodings come from the same video or from different videos. The pose encoder network is trained to produce encoding pairs from the same video that maximize the *entropy* of the scene discriminator. In addition to the adversarial loss, there are reconstruction and similarity losses that exploit notions of temporal slowness in the content features. The model can do forward prediction in latent space simply by fixing the content features and predicting the next pose features conditioned on the fixed content and previous pose features. Questions/comments: - In figure 2, from steps t+2 onward it looks like two h_c vectors are fed into the LSTM instead of an h_c and an h_p. Is this intentional? Based on the model description it seems that predicted h_p should be fed from one state to the next, not h_c. - You may also want to cite “Deep Visual Analogy Making” from NIPS 2015, which uses a factorization of the latent representation into style and content to improve the time horizon of next-frame prediction. The experimental results demonstrate compelling disentangling and impressive long term video prediction across a wide variety of datasets: bouncing digits, NORB, 3D chairs, and KTH videos.

Reviewer 3



This paper propose a model and a loss function that combined together is able to disentangle video frame representations to content and pose. And they shows impressive results in long sequence generation and content and action classification. Some detail comments: -The generated samples looks very sharp, and I am very interested if there were more detailed analyses as what is the cause? Would it be possible to generate samples with wrong pose that still look as sharp? -Table 1: what happens if the concatenated (h_p, h_c) is used for classification? -E3: Shouldn’t it be L(C) ? -E5: Shouldn’t it be L(E_p) and not L(E_c)? -L:219 I’m a bit confused where overfitting is being discussed and referring to Fig 4.3, isn’t the Fig4.3 from Mathieu et al ? Quality: Except some minor probably typos, the paper seems to be technically sound. And has good range of experiments analyzing different aspect of the learned representation. Clarity: It’s well organized, with good review of recent work, model description and results. Minor comment: due to space limit the final part of result section looks a bit cramped, the paper clarity could be improved if larger portion is dedicated to results and analysis section. Originality: The core model and it’s component are simple but the way they have posed the problem and combined those components and the loss function combination is novel. Significance: I think it’s a simple model with very impressive results. Many of the current video generation models degenerate after only few time steps where as this model has sharp samples for longer sequence and could be a work that other people working on video generation could build upon it.